# Dispersion patterns of SARS-CoV-2 variants Gamma, Lambda and Mu in Latin America and the Caribbean

Tiago Gräf [1] ✉, Alexander A. Martinez [2,3,4], Gonzalo Bello [5], Simon Dellicour [6,7], Philippe Lemey [7], Vittoria Colizza [8], Mattia Mazzoli[8], Chiara Poletto[9], Vanessa Leiko Oikawa Cardoso [10], Alexandre Freitas da Silva [11], COVIGEN*, Fernando Couto Motta[12], Paola Cristina Resende [12], Marilda M. Siqueira[12], Leticia Franco[13], Lionel Gresh [13], Jean-Marc Gabastou[13], Angel Rodriguez[13], Andrea Vicari[13], Sylvain Aldighieri [13], Jairo Mendez-Rico[13] & Juliana Almeida Leite [13] ✉

Latin America and Caribbean (LAC) regions were an important epicenter of the COVID-19 pandemic and SARS-CoV-2 evolution. Through the COVID-19 Genomic Surveillance Regional Network (COVIGEN), LAC countries produced an important number of genomic sequencing data that made possible an enhanced SARS-CoV-2 genomic surveillance capacity in the Americas, paving the way for characterization of emerging variants and helping to guide the public health response. In this study we analyzed approximately 300,000 SARS-CoV-2 sequences generated between February 2020 and March 2022 by multiple genomic surveillance efforts in LAC and reconstructed the diffusion patterns of the main variants of concern (VOCs) and of interest (VOIs) possibly originated in the Region. Our phylogenetic analysis revealed that the spread of variants Gamma, Lambda and Mu reflects human mobility patterns due to variations of international air passenger transportation and gradual lifting of social distance measures previously implemented in countries. Our results highlight the potential of genetic data to reconstruct viral spread and unveil preferential routes of viral migrations that are shaped by human mobility patterns.

The COVID-19 pandemic had a significant impact on Latin America and the Caribbean (LAC) countries, with an estimated number of 289 cumulative deaths/100k inhabitants, surpassing Europe and South-East Asia with 240 and 40 reported deaths/100k inhabitants, respectively[1]. Due to its size, ecological diversity and human development inequalities, the COVID-19 burden in LAC countries varied substantially ranging from 4 to 658 deaths/100k hab., although underdiagnosis may have impacted numbers in some countries due to different surveillance strategies. Concomitantly, social distance measures were heterogeneously implemented by governments to contain SARS-CoV-2 spread, both in duration and in strictness[2]. These features, along with the emergence and introduction of new viral lineages in the region, shaped a complex scenario of multiple epidemic waves.

Since the beginning of the pandemic, SARS-CoV-2 genomic surveillance has proven to be a valuable tool to monitor viral spread and the evolution of new variants[3–7]. The timely sequencing of viral genomes and their release in public databases, allowed the identification

A full list of affiliations appears at the end of the paper. *A list of authors and their affiliations appears at the end of the paper. ✉e-mail: tiago.graf@fiocruz.br; leitejul@paho.org

of viral variants that are more transmissible and/or might evade immunity, helping to guide public health response[8]. In the LAC region, the Pan American Health Organization (PAHO), in collaboration with countries' reference and public health laboratories, implemented the COVID-19 Genomic Surveillance Regional Network (COVIGEN), not only for enhancing SARS-CoV-2 genomic sequencing but to generate timely genomic data[9].

Among a myriad of SARS-CoV-2 lineages that evolved during the pandemic, some posed an increased risk to global public health due to significant amino acid substitutions, especially in Spike protein. WHO classified those lineages as Variants of Interest (VOI), when having a regional risk, and Variants of Concern (VOC), when constituting a global concern[10]. In LAC, the emergence of three VOIs (Zeta, Lambda, and Mu) and one VOC (Gamma) demonstrated the regional capacity in genomic sequencing and data analysis. VOI Zeta (P.2, Pango-lineage classification[11]) was firstly detected in Brazil, in October 2020, leading to the resurgence of COVID-19 in the country after the first epidemic wave[12]. Although Zeta was detected world-wide[13], its epidemiological relevance was more evident in Brazil. Zeta's increased growth rate in Brazil was surpassed by the emergence of VOC Gamma (P.1), in December 2020[14,15]. Gamma rapidly spread throughout Brazil[16], causing a second COVID-19 wave, and was detected in more than 80 countries. Still, in December 2020, the first cases of VOI Lambda (C.37) were detected in Peru, triggering a new epidemic wave in the first months of 2021[17]. Lambda has been detected in more than 40 countries. Finally, VOI Mu (B.1.621) was initially identified in January 2021, in Colombia[18], and its spread was also associated with a surge of COVID-19 cases. Almost 60 countries have reported cases of Mu infection[19].

By early 2021, LAC region became an important epicenter of the COVID-19 pandemic. With the emergence of highly transmissible variants, the establishment of COVIGEN, was pivotal to generate the data that allowed variant identification and dispersion tracking. In this study, we analyzed the genetic diversity of the first two years of the SARS-CoV-2 circulation in LAC, focusing on the detailed reconstruction and the diffusion patterns of the three most widespread local variants, Gamma, Lambda and Mu. We used air passenger transportation data to inform phylogeographic models inferring viral flow between locations, accounting for the impact of the different measures of mobility restriction in each country.

## Results

### Sampling overview and molecular diversity

Since the implementation of COVIGEN in March 2020, an increasing number of countries were sampled, with many of them advancing to in country sequencing capacity, resulting in the upscaling of Reference Sequencing Laboratories in the network (Supplementary Table 1). From February 2020 until March 2022, COVIGEN fostered the generation of 126,985 SARS-CoV-2 genomes, sampled in 32 countries/ territories of LAC region. To create a comprehensive view of the viral lineage diversity in the LAC during the pandemic first two years, we complemented the COVIGEN sampling with all SARS-CoV-2 genomes and metadata from the same period and countries as available in Epi-CoV database in GISAID (https://www.gisaid.org/). This approach led to a dataset of 296,286 genomes sampled in 41 countries/territories (Fig. 1).

Among the sampled countries/territories, a highly variable proportion of the total COVID-19 cases were sequenced, from below 0.05% to more than 10%, with a median of 0.6% for the entire region. The top five countries in number of COVID-19 cases (i.e., Brazil, Argentina, Colombia, Mexico, and Peru, in descending order) were responsible for generating ~80% of the sampled genomes, representing 0.4%, 0.1%, 0.3%, 1.0%, and 0.6%, respectively, of their epidemics. To better summarize the diversity of lineages that circulated in the LAC region, we reduced the number of countries/territories ($n = 41$) by aggregating them into 16 countries/regions (hereafter called locations), according to geographical proximity (Fig. 1). For most of these 16 locations, genomic surveillance intensified in the beginning of 2021, after the local emergence of Gamma, Lambda and Mu, and were maintained throughout 2021 and early 2022, due to the introduction and rapid spread of Delta and Omicron, in mid 2021 and late/early 2021/2022, respectively (Supplementary Fig. 1).

Different B.1 descendent lineages (here called "Others") were prevalent during 2020 in the LAC region and different variants dominated the epidemic in 2021 (Fig. 2). Gamma successfully disseminated

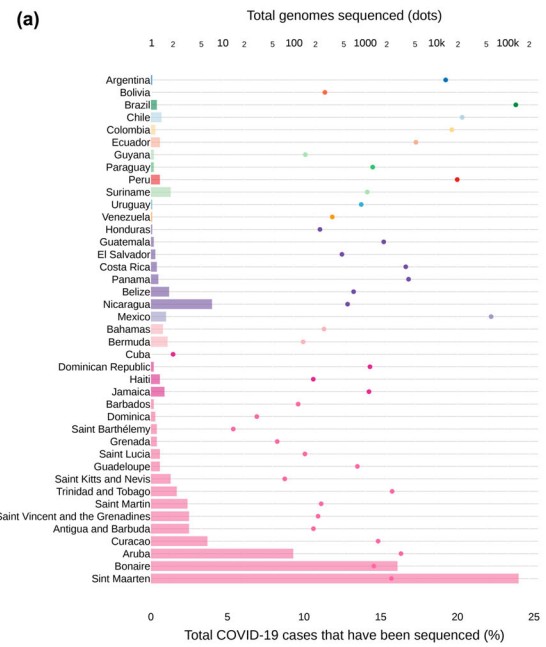

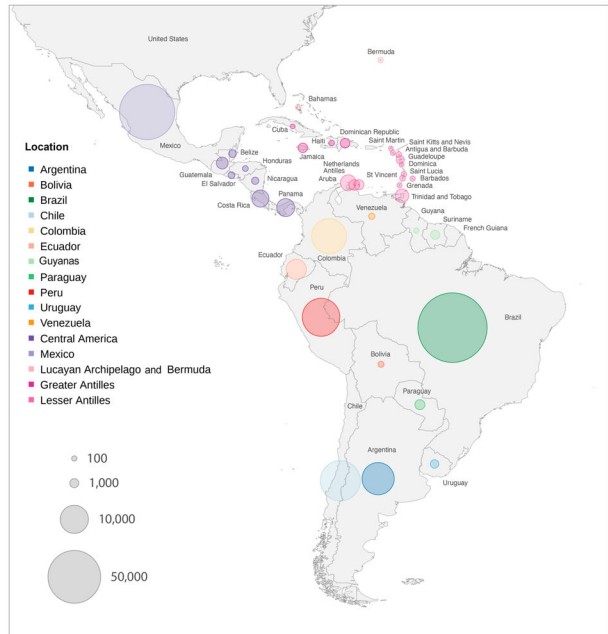

**Fig. 1 | Total number of SARS-CoV-2 genomes per country and proportion of sequenced COVID-19 cases. a** Total number of genomes (top x axis; dots) and proportion of sequenced COVID-19 cases (bottom x axis; bars). **b** Latin America and Caribbean map detailing the sampled countries. Colors are according to the legend, showing the aggregation of countries/territories into locations. Circle sizes are according to the number of genomes generated per country/territory. Source data are provided as a Source data file.

in LAC region with an increasing number of COVID-19 cases attributable to it in almost all analyzed locations since early 2021. Gamma dominated the epidemic in the first half of 2021 in Brazil, Bolivia, Paraguay, Uruguay, the Guianas and the Lesser Antilles. In the first half of 2021, Lambda dominated the epidemic in Peru, while Mu dominated the epidemic in Colombia. Some countries displayed a co-circulation of variants Gamma and Lambda (Argentina and Chile), Gamma and Mu (Venezuela, Mexico, Central America, and the Greater Antilles), or all three variants (Ecuador) (Figs. 2 and 3a). Delta was the dominant variant in all countries in the second half of 2021. Except for Venezuela and the Guianas, in most South American countries, Delta did not lead to a resurgence of COVID-19 cases as was observed in Mexico, Central America, and the Caribbean (Fig. 2). In contrast to Delta, Omicron introduction and spread in the LAC region translated into a steep increase in the number of cases in late/early 2021/2022 for all locations sampled in this study, although for some locations genomic data on the circulating virus in this period was not available.

## Phylogeography of the Gamma, Lambda and Mu variants in Latin American and Caribbean

To study the patterns of viral migration between countries in LAC region, we performed Bayesian phylogeographic analyses for the three most important VOC/VOIs that emerged in the region. Genomes classified as Gamma (P.1 + P.1.*), Lambda (C.37 + C.37.*) or Mu (B.1.621 + B.1.621.*) and with >29,000 nt and <5% of Ns were selected (Supplementary Table 1). After removing identical sequences, we downsampled the dataset proportionally to the attributable number of COVID-19 cases per variant in each location (see "Methods" for details). This approach resulted in a dataset of 2299 genomes for Gamma, 1903 genomes for Lambda and 2586 genomes for Mu. To reconstruct the potential pattern of viral spread, we used international air passenger

transportation data to inform the phylogeographic model in an integrated framework. Variation in the total air passenger transportation among locations was also integrated in the model to accommodate the different pandemic containment strategies implemented in each country (see "Methods" for details).

Bayesian phylogeography estimated that the ancestral location at the root of the Gamma, Lambda and Mu phylogenies, were Brazil (0.99, posterior probability [pp]), Peru (0.99, pp), and Colombia (0.68, pp), respectively, from where these variants disseminated regionally (Fig. 3a). The dynamics between viral flow (estimated from the phylogenies) and the number of COVID-19 cases attributable to each variant is remarkably well synchronized (Fig. 3a). This is clearly observed in Argentina and Chile, where the number of Gamma and Lambda cases rises soon after the most intense period of variant inflow. In these countries, as the Gamma and Lambda epidemics increase locally, viral outflow increased as well. A similar pattern is observed for the variants Mu in the Greater Antilles and Gamma in Venezuela. The latter country seems to have made an early (January and February 2021) contribution to the spread of Gamma. International spread was inferred to have occurred mainly between December 2020 and March 2021 for Gamma and Lambda and between April and June 2021 for Mu (Fig. 3a and Supplementary Fig. 2).

Our analyses support a significant association between air passenger transportation data and among-location viral flow for Gamma [β = 0.86 (0.60, 1.11, 95%HPD)] and Lambda [β = 1.11 (0.75, 1.48, 95% HPD)], but less evident for Mu [β = 0.28 [0.07, 0.47, 95%HPD)]. The most intensive period of Gamma and Lambda international spread (December 2020 to March 2021) coincided with a transient increase in the number of flights associated with holiday season, particularly in South America (Fig. 3b). The total viral flow for Gamma and Lambda between sampled locations significantly decreased after travel

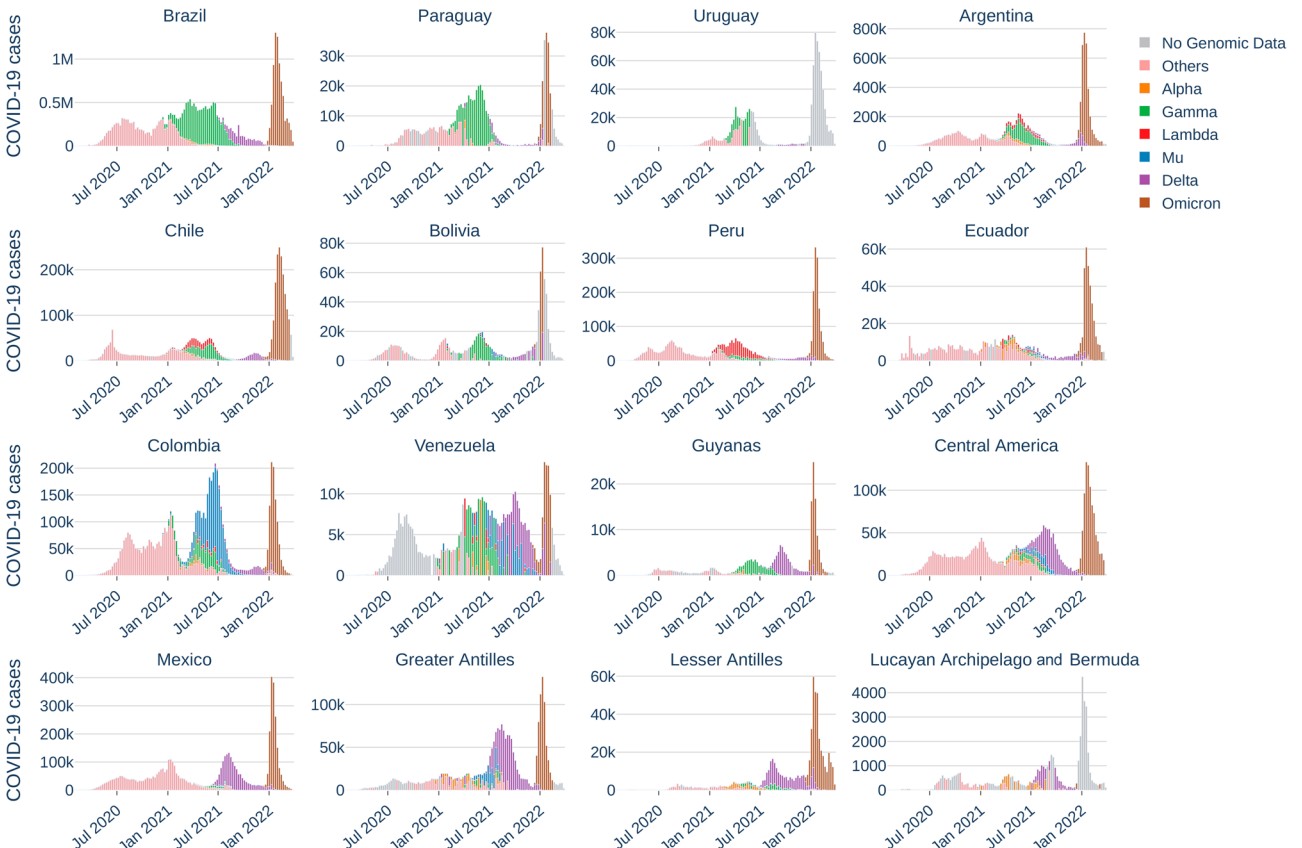

**Fig. 2 | Weekly number of COVID-19 cases attributable to the main viral lineages circulating between February 2020 and March 2022 in each studied location.** Source data are provided as a Source data file.

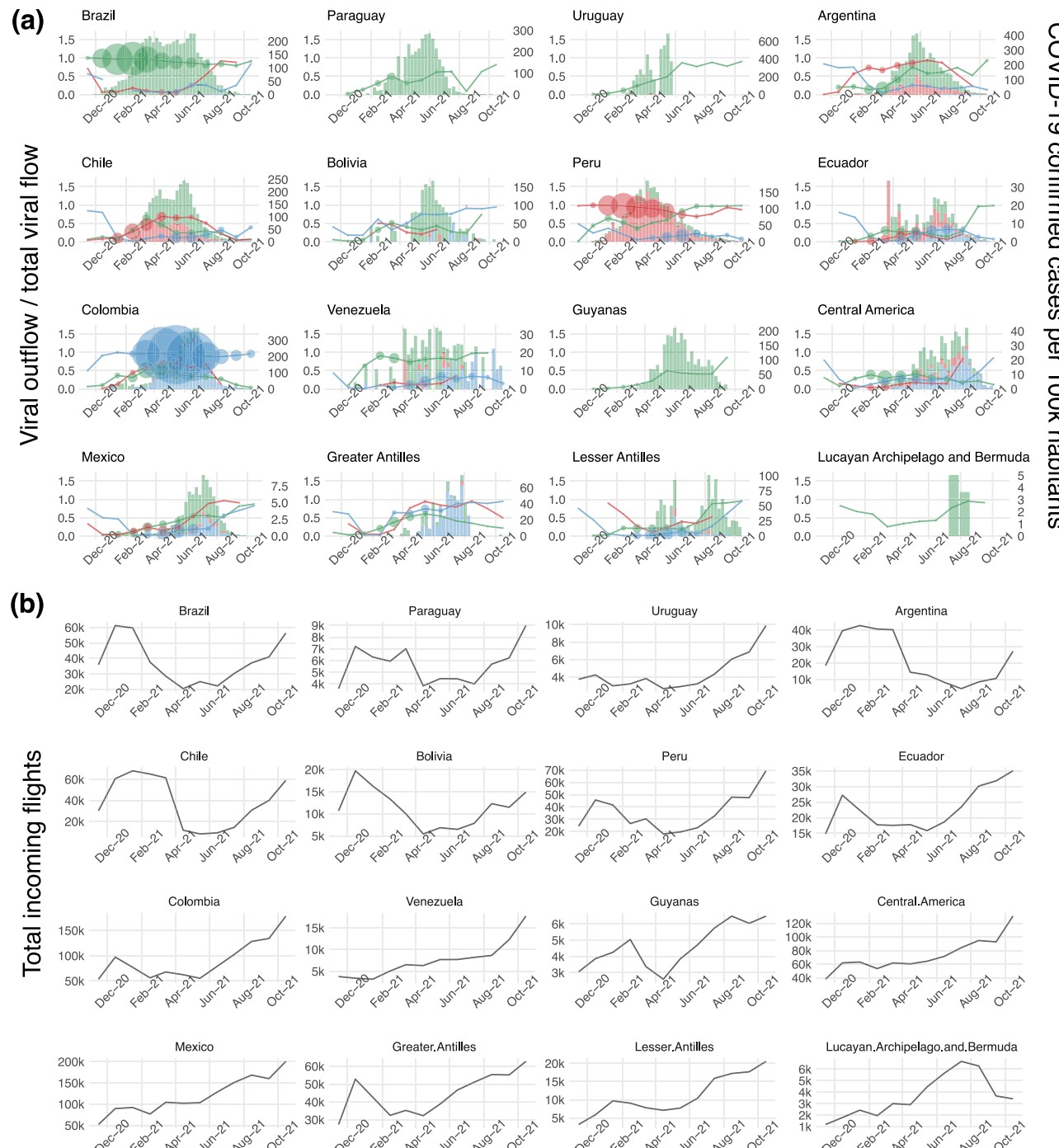

**Fig. 3 | SARS-CoV-2 variants and air passenger flow among Latin America and Caribbean locations. a** Ratio of variant specific viral outflow over total viral flow and number of COVID-19 confirmed cases through time. Left y-axis, number of viral exportations (outflow) over total viral flow (viral inflow plus viral outflow) per location. Size of the circles are proportional to the total viral flow in each location. Right y-axis, weekly number of COVID-19 confirmed cases attributable to variants Gamma (green), Lambda (red) and Mu (blue). **b** Monthly number of incoming flights by studied location. Total number of flights considers only flights between countries in Latin America and the Caribbean. Since incoming and outgoing flights are almost symmetrical, we show only incoming flights. Source data are provided as a Source data file.

restrictions (April to June 2021), even though the number of COVID-19 cases attributable to these variants were still high, especially in Brazil for Gamma variant (Fig. 3a). The air traffic in Venezuela, Mexico, Central America, and the Caribbean remained relatively limited between December 2020 and March 2021, and this might have reduced the chance of exportation of Gamma and Lambda variants to those locations. On the other hand, the main bulk of Mu variant outflow from Colombia (April to June 2021) coincided with the period of

most intense international air travel restrictions in Colombia and many other countries in South America. This might explain the lower predictive value of air traffic when modeling the Mu diffusion process and land and water-based types of transport might have played an important role, especially for the spread of Mu to Venezuela/Ecuador and Central America/Caribbean, respectively.

Besides the main role of the countries where Gamma, Lambda and Mu emerged (i.e., Brazil, Peru, and Colombia, respectively), our

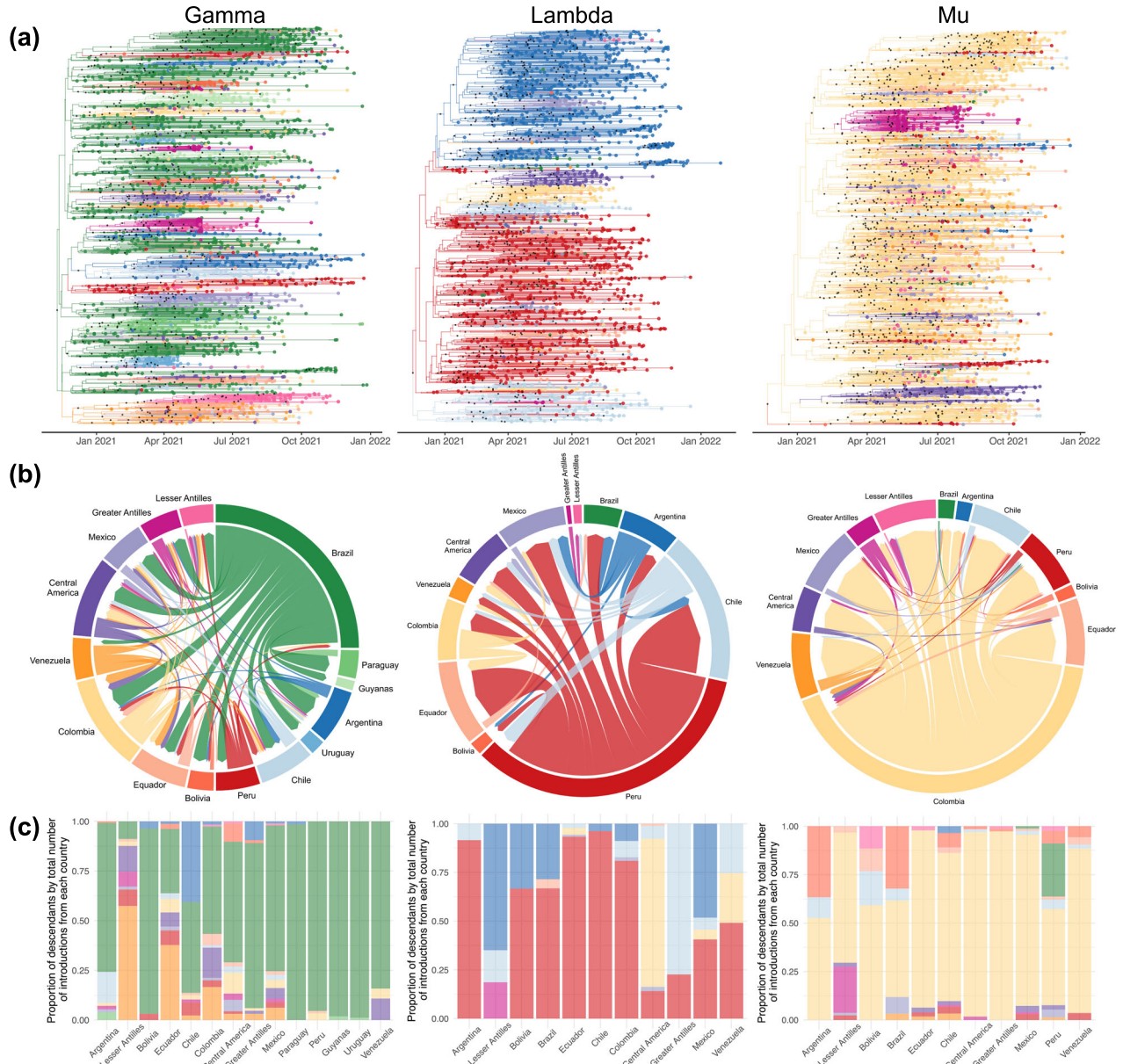

**Fig. 4 | Spatio-temporal viral spread of SARS-CoV-2 variants Gamma, Lambda, and Mu. a** Time-scaled maximum clade credibility tree with ancestral node locations inferred by Bayesian Phylogeography. Nodes with posterior probability higher than 0.90 are indicated with a black dot. **b** Circular migration flow plots for all sampled locations based on the number of Markov jumps. **c** Proportion of descendant lineages by total number of introductions from each location. The colors of (**a**) and (**c**) are according to the circular migration flow plots (**b**). Source data are provided as a Source Data file.

study unveiled complex migration pattern where many locations contributed for the regional dissemination (Fig. 4a). The exchange of viral lineages among locations was higher for Gamma than for the other variants, and many locations played a role on international transmission. Argentina, Colombia, Peru, Venezuela, and Central America were estimated as secondary locations of Gamma spread when considering the total number of jumps (Fig. 4b and Supplementary Fig 2). This is observed at a lesser degree for Lambda, for which Argentina, Chile, and Colombia were estimated as locations where secondary routes disseminations were identified. Meanwhile, Bolivia and the Greater Antilles appeared to be important secondary routes for Mu (Fig. 4b and Supplementary Fig 2). Other countries from South America (Guyana, Suriname, Paraguay, and Uruguay) and the Caribbean (Lesser Antilles) did not appear to play a relevant role for regional dissemination of Gamma, Lambda or Mu.

Since not every lineage introduction will necessarily spread successfully within a new location, we measured the proportion of descendants by the total number of introductions by each location (Fig. 4c). This analysis produced a clearer picture of the role of each location in seeding viral lineages. Besides the expected impact of Brazil, Peru, and Colombia, other countries were route for viral spread through the region. Viral outflow from Venezuela was estimated to contribute to 57% of the Gamma lineages circulating in the Lesser Antilles and 38% in Ecuador. Around 32% of the Gamma lineages in Colombia moved from Venezuela or Central America, and 41% of the Gamma sequences in Chile were estimated to be linked to Argentina. When reconstructing the Lambda spread, our analysis estimated that 76% of the viruses circulating in Central America are related to Colombia, 65% and 48% of Lambda in the Lesser Antilles and Mexico, respectively, are linked to Argentina, and 77% in the Greater Antilles are related to Chile. Finally, the spread of 37% and 32% of the Mu lineages

circulating in Argentina and Brazil, respectively, were estimated to have moved from Bolivia, and the Greater Antilles was responsible for 24% of the Mu lineages in the Lesser Antilles.

## Discussion

Previously limited to highly specialized laboratories, SARS-CoV-2 genomic sequencing had a critical role in the surveillance of viral lineages during the pandemic. In this sense, COVIGEN played a central role in strengthening localized genomic surveillance capacity through access to equipment, critical reagents, and training of laboratory personnel, or by fostering the use of external sequencing refence laboratories based in Mexico, Brazil, Chile, Colombia, Costa-Rica, Panama, Trinidad y Tobago, and USA[9]. In the first two years of the pandemic, approximately 43% of all SARS-CoV-2 genomes from the LAC region were generated in laboratories being supported by PAHO, showing the importance of international collaboration networking.

With the pandemic emergence, an unprecedented number of genomes were generated, demanding the development of new and the improvement of existing methods for genetic data analyses, from viral lineage classification schemes and data storage/sharing solutions[11,20-22], to complex inference of virus evolution and migration[4,23]. In the current study, we have used state-of-the-art methods to investigate the spread of SARS-CoV-2 variants Gamma, Lambda and Mu in the LAC region. Specifically, to account for the effect of divergent travel restrictions measures across the territory, we have implemented a recently developed method that incorporates variation in mobility over time in the phylogeographic model[4]. This allowed us to unveil patterns of viral migration that are related to epidemic scenarios that follows intense periods of viral inflow. Interestingly, our study revealed the role of Colombia and Venezuela connecting South America with Central America and Caribbean. Venezuela was estimated to have an early role in Gamma dissemination, mainly to the Lesser Antilles, and this might be related to its geographical proximity to the Brazilian Amazon region, where Gamma has first emerged[15]. Colombia was the main origin of Lambda lineages circulating in Central America, which might be related to the dominance of Lambda in the country's Pacific region[24], and the whole Mu epidemic, originating from Colombia, was much more effective in spreading towards the north than to the south. Argentina and Chile were also observed as important routes of Lambda spreading towards the Caribbean and Mexico, which agreed with previous studies investigating the Lambda epidemic wave in these two countries[25,26]. Another identified pattern that emerged within South America is the preferential viral migration among neighboring countries located in the western side (Colombia, Ecuador, and Peru) or the eastern side (Brazil, Argentina, Paraguay, and Uruguay) of the Andes; while Chile, Bolivia and Venezuela seem to be strongly connected with both sides.

Although all possible country points of entry considered by the International Health Regulation, IHR (i.e., international airports, ports, and ground crossing) might have played a role in the variants' dispersion, in this study air passenger transportation data was estimated as an important predictor of viral diffusion in the LAC region. In Europe, Google mobility data (encompassing both air and land-based transport) was shown to offer a stronger correlation with SARS-CoV-2 spread than international air travel alone[4]. Although we could not access Google mobility data for the LAC region, the association between air travel and virus dispersion is in line with the big territory and large distances between main cities. It is important to note that three of the top five air transportation hubs in Latin America (Guarulhos International Airport in Brazil, Jorge Chávez International Airport in Peru, and El Dorado International Airport in Colombia)[24] are located in the countries where Gamma, Lambda and Mu firstly originated. Further, the air connectivity index[27], which measure the degree of integration of a country into the global air transport network, suggests that VOCs and VOIs emerging in Brazil, Colombia, Peru, Mexico, or Panama may have a greater chance of spreading. However, data from Mexico suggest that travel restrictions are not the only effective control measure. Mexico, which did not implement strict travel restrictions and had a strong genomic surveillance system (with one of COVIGEN Reference Sequencing Laboratory), was not estimated as contributing to VOC/VOI disseminations. These findings show that other measures as social distancing and use of mask were also effective to control dispersion.

Although the total number of viral transition events between sampled locations significantly decreased upon imposing travel restrictions, the virus diffusion across countries was not completely blocked. This is mainly due to two factors. First, in absence of complete travel bans (i.e., zero air traffic), importations still occur, despite at a lower rate, due to the residual air travel fluxes coupled with the large case growth in the origin country[28-30]. Second, short-distances dispersal through land and water might have also played an important role in the spreading of SARS-CoV-2 variants among LAC regions. This could be particularly relevant for Mu dissemination outside Colombia during a period of intense international air travel restrictions in the country. Consistent with this pattern, air passenger transportation data had a smaller predictive value to reconstruct the spread of Mu. Previous studies already pointed to the relevance of land-based human movements for the spread of SARS-CoV-2 across the Brazilian-Uruguayan, Brazilian-Venezuelan, and Colombian-Venezuelan borders[31-34], suggesting that short-distance travels by land could also be a significant contributor to the spread of SARS-CoV-2. Moreover, maritime transport across major hub ports in the Caribbean could also be an alternative route of viral dissemination between northern South America, Central America, and the Caribbean. The presence of natural barriers, such as the Darien Gap (a dense forest in the border of Colombia and Panamá), hinder the land transport between South and Central America, highlighting the importance of maritime transport. However, a direct assessment of alternative (besides airplanes) modes of transportation is challenging due to the lack of detailed data on its routes. In agreement with the results presented here, a study investigating the initial chikungunya virus spread in the Caribbean islands found that spatial transmission was better described by geographic proximity than by air transportation fluxes, which is very likely explained by the movement of individuals by boat[35].

Studies have shown that VOI Mu is less susceptible to antibody neutralization, from both convalescent and vaccinated sera, when compared to the VOCs Alpha, Beta, Gamma and Delta[36,37]. Nevertheless, such high immune evasiveness was not enough to promote a successful spread of Mu on the LAC region. Several factors may explain that. First, Gamma and Lambda spread during the holidays and school summer vacation (mid-December to February) in South America, when family traveling is frequent and the impact of summer travel on SARS-CoV-2 dissemination was also demonstrated in Europe[4,38,39]. COVID-19 cases attributable to Mu variant, by contrast, started to increase in Colombia after summer holidays (March 2021) and during a period of intense international air travel restrictions. Second, when Mu emerged, Gamma and Lambda were already established in many countries and local social distance measures to control the increasing number of COVID-19 cases might have decreased the probability of Mu to establish successful community transmission. Third, Mu faced competition from Gamma/Lambda in South American countries and from Delta in Mexico, Central America, and the Caribbean. Delta variant transmissibility was estimated as being the highest among the variants that emerged before it[36,40] and this feature might have offered an advantage to Delta over Mu where they co-circulated, as also suggested by[41]. Fourth, dissemination of Mu may have been limited by the "hybrid immunity" generated by the combination of natural infection and vaccination in LAC region[42-44]. Indeed, a previous study shows the loss of correlation between population mobility and Gamma and Lambda effective reproduction number in countries from South

American Southern cone since middle 2021[45]. This was associated with the increasing population hybrid immunity, and we may speculate that during the period of most intense international spread of Mu (April-June 2021), populations from the South America may have already achieved the conditional herd immunity threshold to contain the dissemination of regional SARS-CoV-2 variants, including Mu.

Our study has some caveats and the results presented here should be interpreted considering that. Firstly, the uneven sampling between studied regions may impact the phylogeographic reconstructions, particularly from Guianas, a potential intermediate region for viral dissemination between Lesser Antilles and South America. To mitigate this bias, we have sub-sampled genomes based on variant attributable case counts in each location, even though we were limited by the number of genomes publicly available. Secondly, although we have focused on the VOC/VOI that emerged in the LAC, the exclusion of countries from North America and Europe might preclude the identification of transition events between such countries and the LAC. For example, USA and Mexico were shown to have a related Gamma epidemic, with genomes clustering together, especially for the P.1.17 Gamma sublineage[46]. However, the exclusion of countries from Europe and North America probably did not have a great impact on the inferred spread patterns of Lambda and Mu due to the very low prevalence of those variants outside South America. Thirdly, it was not possible to obtain and analyze data from all the different country points of entry according to the IHR; nevertheless, the international air transportation might represent a critical and important source to estimate international movement patterns.

In summary, our study revealed that SARS-CoV-2 variants Gamma, Lambda and Mu dissemination in LAC region was facilitated by corridors that had Colombia and Venezuela as possible major staging-posts between South America, Central America and the Caribbean. The geographic dispersal of Gamma and Lambda that arose in Brazil and Peru, respectively, and mostly spread during the holidays and school summer vacations between December 2020 and February 2021, was mostly explained by international air passenger transportation. The variant Mu arose in Colombia and mainly spread between April and June 2021, after the summer vacations and during a period of intense international air travel restrictions in Colombia that might explain the more restricted regional dissemination of this variant and the lower predictive value of air traffic when modeling the Mu diffusion process. These findings help to understand the impact of SARS-CoV-2 variants in the COVID-19 epidemic dynamics in LAC region and highlight viral genomic surveillance as a powerful tool not only to rapidly identify genetic variants with relevant mutations, but also to infer dispersion patterns that might explain outbreaks and epidemics. In this sense, PAHO has lunched the Strategy on Regional Genomic Surveillance for Epidemic and Pandemic Preparedness and Response[47], aiming to strength the genomic surveillance in the Americas to other pathogens, fostering timely release of the genomic data and the fully integration to surveillance systems, which will be ultimately useful during implementation of public health control measures and decision making.

## Methods

### Strengthening of genomic surveillance capacity through COVID-19 Genomic Surveillance Regional Network (COVIGEN)

As a collaborative networking among PAHO, national authorities, and national public health laboratories, COVIGEN was implemented for strengthening timely SARS-CoV-2 genomic surveillance, facilitating streamlined logistics, procurement and distribution of sequencing reagents, in-country and sub-regional trainings in genomic sequencing and bioinformatics, and for guidance on official notifications through international health regulations. The heterogenic network was based on Nanopore and Illumina technologies and was composed by countries that perform in-house sequencing or external sequencing, held on reference sequencing laboratories.

### SARS-CoV-2 sequencing strategies and ethical aspects

Sequencing was not conducted as part of this study, but the study instead used sequences previously generated by COVIGEN and other genomic surveillance initiatives in Latin America and the Caribbean. The COVIGEN network, selected samples from routine surveillance based on representativeness and virologic criteria[48]. According to national protocols, samples collected in the context of the national epidemiological surveillance, do not require informed concern if they are going to be use only to detect and characterize the possible pathogen involved in an outbreak or the pathogen under surveillance. The identity of the patients sampled in this context remained anonymous and samples were not used to detect or research for human biological markers or any other different to the surveillance purposes. Sequences generated by countries in COVIGEN network were obtained through next-generation sequencing (NGS) methodology, based on Illumina or Nanopore technologies, implemented in countries reference laboratories or through external sequencing at one of the COVIGEN Regional Sequencing Laboratory. Nanopore sequencing was performed following a previously published protocol[49]. Illumina sequencing was performed with the Illumina COVIDseq Test, according to manufacturer's instructions, and clustered with MiSeq Reagent Kit v2 (500-cycles - MS-102-2003) on 2 × 250 cycles (in-house protocols) or 2 × 150 cycles (MS-103-1002) paired-end runs. All sequencing data was collected using the Illumina MiSeq sequencing platforms and MiSeq Control software v2.6.2.1. Details are provided in ref. 15. Sequences were made available in a timely manner, into the EpiCov database in GISAID platform[21].

### Sequence data and subsampling

Initially, all SARS-CoV-2 genomic metadata from all countries and territories in Latin America and the Caribbean (LAC), collected up to 31 March 2022, were retrieved from the EpiCoV database in GISAID (https://www.gisaid.org/) as accessed on 13 April 2022. Sequences without a complete collection date were excluded and variants different from Alpha (B.1.1.7 + Q.*), Gamma (P.1 + P.1.*), Delta (B.1.617.2 + AY.*), Lambda (C.37 + C.37.*), Mu (B.1.621 + B.1.621.*) and Omicron (B.1.1.529 + BA.*) were aggregated as "Others". To better summarize the diversity of variants in the LAC region, we reduced the number of countries/territories ($n = 41$) by aggregating them into 16 countries/regions (hereafter called locations), according to geographical proximity. This dataset was used to produce the sampling visualizations and SARS-CoV-2 variants diversity through time for each sampled location.

From this dataset, we then selected only Gamma, Lambda and Mu entries whose genome sequences had >29,000 nt and <5% of Ns and those were retrieved from the GISAID complete alignment. To reduce computational burden and sampling disparities among locations, we removed duplicated sequences for each location with Seqkit program[50] and sub-sampled each variant dataset proportionally to the cumulative number of COVID-19 cases attributable to Gamma, Lambda and Mu in each location. An arbitrary number of 1 sequence/10,000 Gamma attributable COVID-19 cases and 1 sequence/1000 Lambda or Mu attributable COVID-19 cases were selected, with a minimum number of 100 sequences (or the maximum available when less than that) per location. To maximize the temporal coverage, sequences were grouped by epidemiological week and sampled as evenly as possible in each location. This approach resulted in alignments of 2417 Gamma genomes, 1992 Lambda genomes and 2618 Mu genomes, which were visually inspected in AliView[51].

Maximum likelihood (ML) phylogenetic trees were inferred in IQ-TREE v2.1.2[52] under the general time-reversible (GTR) model of nucleotide substitution with four categories of gamma-distributed rate variation among sites (G4). To maximize the temporal signal of the Gamma, Lambda and Mu datasets, a regression analysis of the root-to-tip divergence against tip sampling time of the ML phylogenetic trees was performed in TempEst[53] and outlier sequences that deviate more

than 1.5 interquartile ranges from the root-to-tip regression line were excluded. Final datasets contained 2299 genomes for Gamma, 1903 genomes for Lambda, and 2586 genomes for Mu.

## Bayesian phylogeography

Time-scaled phylogenetic trees were inferred in BEAST 1.10[54] using the GTR + G4 nucleotide substitution model, a strict molecular clock with a fixed rate of $7.5 \times 10^{-4}$ substitutions/site/year, and the Bayesian skygrid coalescent model to infer effective population size trajectories in grids encompassing a two-week interval[55]. We made use of the Hamiltonian Monte Carlo (HMC) gradient-based sampler to efficiently estimate the skygrid model parameters[56]. Viral migration patterns between sampled locations were modeled by treating discrete location states as an evolving character and the transition between states was estimated with a continuous-time Markov chain (CTMC)[57]. Viral transition rates between sampled locations were modeled as a log-linear function of the population mobility among locations[58]. This is implemented in BEAST 1.10 as an integrated framework to reconstruct phylogenetic history, ancestral location states and the contribution of the mobility data (the β coefficient) as a covariate of the viral transition rates estimated from the genetic data. We used air passenger transportation data based on the number of origin-destination tickets, obtained from the International Air Transport Association (http://www.iata.org), as a predictor for people's mobility.

Initially, we set-up a time-homogeneous model where a single viral transition matrix between locations is estimated, summarizing the whole variant evolutionary history, and the contribution of the mobility data is assessed as the sum of all air passengers between each pair of locations in the analyzed time. This approach does not take into account the time variability of mobility between locations but can efficiently be used in an integrated framework to jointly estimate the phylogenetic trees and viral migration patterns. The BEAGLE library v.3[59] was used to increase computational performance and multiple independent Markov chain Monte Carlo (MCMC) runs were performed and later combined to ensure that all continuous parameters had an effective sample size (ESS) > 200, as visualized in Tracer v1.7[60].

To accommodate the severe mobility restrictions implemented during the pandemic, a time-inhomogeneous model was constructed where we specified arbitrary time intervals over the evolutionary history (epochs) and applied different model parameters to them[61]. In the time-inhomogeneous model we implemented monthly air-passenger flow matrices and estimated the contribution of all matrices as a single model predictor. To reduce the computational burden associated with the time-inhomogeneous model, it was fitted to a set of posterior trees estimated from the full-time-homogeneous model. This eliminated the necessity of the simultaneous tree estimation and only spatial-diffusion-related parameters were inferred, reducing the parameter space and facilitating MCMC sampling.

## Posterior summarization and visualization

After removing 10% of burn-in, transitions to (inflow) and from (outflow) each sampled location were summarized based on the Markov jump estimation[62]. Each location's total viral flow was calculated as the sum of the inflow and outflow. The BEAST tool TreeMarkovJumpHistoryAnalyzer (available at: https://github.com/beast-dev/beast-mcmc) was used to extract location transition history from the posterior tree distribution and the number of descendants of the same state each transition resulted. Plots were generated in the "ggplot2" package[63]. Maximum clade credibility (MCC) trees were summarized with the TreeAnnotator, and plotted with "treeio"[64] and "ggtree"[65] package. Finally, circular migration plots were generated with the Markov jump data in the "circlize"[66]. COVID-19 cases and deaths data in Latin America and the Caribbean were assembled by PAHO and kindly provided to this study.

## Reporting summary

Further information on research design is available in the Nature Portfolio Reporting Summary linked to this article.

## Data availability

SARS-CoV-2 genomes used in these analyses were downloaded from EpiCoV database in GISAID (https://www.gisaid.org/) and are available at https://gisaid.org under the EPI_SET_230926ex code locator. Proprietary air travel data are commercially available from the International Air Transport Association (https://www.iata.org/) databases and cannot be publicly shared. Source data are provided with this paper.

## Code availability

All BEAST xml files used in this study are available at https://github.com/viromol/SC2_LAC-region_phylogeography.git (https://zenodo.org/doi/10.5281/zenodo.10594221).

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

## Acknowledgements

We are grateful for the work of all National Influenza Centers, public health laboratories and other institutions contributing for timely generating genomic sequences for COVID-19 pandemic response. We would like to thank GISAID and all the submitters of the database, as well as to the COVIGEN Reference Sequencing Laboratories (Centers for Disease Control and Prevention, Atlanta, United States of America; Instituto Costarricense de Investigación y Enseñanza en Nutrición y Salud, Cartago, Costa Rica; Instituto de Diagnóstico y Referencia Epidemiologicos, Mexico City, Mexico; Instituto de Salud Pública de Chile, Santiago, Chile; Instituto Nacional de Salud, Bogota, Colombia; Gorgas Memorial Institute for Health Studies, Panama City, Panama; Laboratório de Vírus Respiratórios, Exantemáticos, Enterovírus e Emergências Virais, Instituto Oswaldo Cruz, Fundação Oswaldo Cruz, Rio de Janeiro, Brazil; The University of the West Indies - St. Augustine, St. Augustine, Trinidad and Tobago) for all their contribution for the network implementation as well as on making possible external sequencing for the network. We are also grateful for the support of personnel at PAHO headquarters and PAHO/WHO Country Offices for the technical cooperation provided to the COVID-19 response. We acknowledge USA-CDC and USA-HSS for direct funds contribution to PAHO. This work was also supported by the Foreign, Commonwealth & Development Office and Wellcome (grant number: 223610/B/21/Z). We are also grateful to all the donors to WHO for additional contributions. SD also acknowledges support from the *Fonds National de la Recherche Scientifique* (F.R.S.-FNRS, Belgium; grant n°F.4515.22) and from the Research Foundation — Flanders (*Fonds voor Wetenschappelijk Onderzoek - Vlaanderen*, FWO, Belgium; grant n°G098321N). S.D., P.L., and V.C. acknowledge support from the European Union Horizon 2020 project MOOD (grant agreement n°874850). P.L. acknowledges support from the European Research Council under the European Union's Horizon 2020 research and innovation program (grant agreement no. 725422 - ReservoirDOCS) and the National Institutes of Health grant R01 AI153044. G.B. is funded by CNPq (Grant number 304883/2020-4) and FAPERJ (Grant number E-26/202.896/

2018). M.M.S. acknowledges funding from USA-CDC, NVAP-UKHSA, DECIT - MoH Brazil and CNPQ (Grant number 402457/2020-0).

## Author contributions

T.G.: Conceptualization, methodology, formal analysis, data curation, writing—original draft. A.A.M.: Conceptualization, investigation, data curation, writing—review & editing. G.B.: Conceptualization, writing—original draft. S.D. and P.L.: Methodology, formal analysis, software, writing—review & editing. V.C., C.P. and M.M.: Investigation, data curation, writing—eview & editing. T.G., V.L.O.C., S.D. and P.L.: Visualization. AFS: Formal analysis. PCR, MMS: Investigation, data curation, writing—review & editing. L.F., L.G., J.M.G., A.R., A.V. and S.A.: Project administration, writing—review & editing. F.C.M., J.M.R. and J.A.L.: Conceptualization, supervision, funding acquisition, writing—review & editing.

## Competing interests

The authors declare no competing interests.

## Additional information

[1]Laboratório de Virologia Molecular, Instituto Carlos Chagas, Fundação Oswaldo Cruz, Curitiba, Brazil. [2]Gorgas Memorial Institute for Health Studies, Panama City, Panama. [3]National Research System (SNI), National Secretary of Research, Technology and Innovation (SENACYT), Panama City, Panama. [4]Department of Microbiology and Immunology, University of Panama, Panama City, Panama. [5]Laboratório de Arbovírus e Vírus Hemorrágicos, Instituto Oswaldo Cruz, FIOCRUZ, Rio de Janeiro, Brazil. [6]Spatial Epidemiology Lab (SpELL), Université Libre de Bruxelles, CP160/12, 50 av. FD Roosevelt, Bruxelles, Belgium. [7]Department of Microbiology, Immunology and Transplantation, Rega Institute, Laboratory for Clinical and Epidemiological Virology, KU Leuven, University of Leuven, Leuven, Belgium. [8]Sorbonne Université, INSERM, Institut Pierre Louis d'Épidémiologie et de Santé Publique (IPLESP), Paris, France. [9]Department of Molecular Medicine, University of Padova, 35121 Padova, Italy. [10]Laboratório de Enfermidades Infecciosas Transmitidas por Vetores, Instituto Gonçalo Moniz, FIOCRUZ-Bahia, Salvador, Brazil. [11]Núcleo de Bioinformática, Instituto Aggeu Magalhães, Fundação

Oswaldo Cruz, Pernambuco, Brazil. [12]Laboratório de Vírus Respiratórios, Exantemáticos, Enterovírus e Emergências Virais, Instituto Oswaldo Cruz, Fundação Oswaldo Cruz, Rio de Janeiro, Brazil. [13]Infectious Hazards Management Unit, Health Emergencies Department, Pan American Health Organization, Washington D.C., USA. ✉e-mail: tiago.graf@fiocruz.br; leitejul@paho.org

## COVIGEN

Juliana Almeida Leite ⑩ [13]✉, Jairo Mendez-Rico[13], Andrea Vicari[13], Elsa Baumeister[14], Josefina Campos[14], Andrea Pontoriero[14], Indira Martin[15], Kirvina Ferguson[15], Draven Johnson[15], Songee Beckles[16], Kasandra Forde[16], Aldo Sosa[17], Roberto Melendez[17], Roxana Loayza[18], Cinthia Avila[18], Evelin Esther Fortun Fernández[19], Carol Jessica Guzman Otazo[19], Marilda Mendonça Siqueira[12], Fernando Couto Motta[12], Paola Cristina Resende ⑩ [12], Katia Corrêa de Oliveira Santos[20], Adriano Abbud[20], Mirleide Cordeiro dos Santos[21], Jessylene de Almeida Ferreira[21], Rodrigo Fasce[22], Jorge Fernandes[22], Sergio Gómez Rangel[23], Marcela Maria Mercado[23], Dioselina Pelaez[23], Claudio Soto-Garita[24], Estela Cordero-Laurent[24], Francisco Duarte-Martínez[24], Hebleen Brenes[24], Isaac Miguel Sanchez[25], Yvonne Imbert[25], Alfredo Bruno Caicedo[26], Domenica Joseth de Mora Coloma[26], Dalia Xochitl Sandoval López[27], Denis Gerson Jovel Alvarado[27], Deny Lisset Martínez Morán[27], Claudia Pacheco[28], Linda Mendoza[28], Joyce Whyte-Chin[29], Mustapha Abdul-Kadir[29], Jacques Boncy[30], Ito Journel[30], Mitzi Castro Paz[31], Sofia Carolina Alvarado[31], Soany Avilez[31], Michelle Brown[32], Caludia Elena Wong Arambula[33], Ernesto Ramírez González[33], Alexander A. Martinez[2], Claudia M. Gonzalez[2], Brechla Moreno Arevalo[2], Danilo Franco[2], Sandra Lopez-Verges[2], Juan Miguel Pascale[2], Cynthia Vazquez[34], Sandra Gonzalez[34], Nancy Rojas Serrano[35], Carlos Patricio Padilla Rojas[35], Phyllis Pinas[36], Navienda Asebeh[36], Christine V. F. Carrington[37], Nikita S. D. Sahadeo[37], Hector Chiparelli[38], Natalia Goñi[38], Lieska Rodriguez[39] & Pierina D'Angelo[39]

[14]Instituto Nacional de Enfermedades Infecciosas-ANLIS C.G.Malbran, Buenos Aires, Argentina. [15]Ministry of Health Reference Laboratory, Nassau, Bahamas. [16]Best-Dos Santos Public Health Laboratory, Bridgetown, Barbados. [17]Medical Laboratory Services of the Ministry of Health and Wellness, Belmopan, Belize. [18]Centro Nacional de Enfermedades Tropicales, Santa Cruz, Bolivia. [19]Instituto Nacional de Laboratorios de Salud, La Paz, Bolivia. [20]Instituto Adolfo Lutz, São Paulo, Brazil. [21]Instituto Evandro Chagas, Ananindeua, Para, Brazil. [22]Instituto de Salud Pública de Chile, Santiago, Chile. [23]Instituto Nacional de Salud, Bogota, Colombia. [24]Instituto Costarricense de Investigación y Enseñanza en Nutrición y Salud, Cartago, Costa Rica. [25]Laboratorio Nacional de Salud Publica Dr. Defilló, Santo Domingo, Dominican Republic. [26]Instituto Nacional de Investigación en Salud Pública, Guayaquil, Ecuador. [27]Laboratorio Nacional de Salud Publica "Dr. Max Bloch", San Salvador, El Salvador. [28]Laboratorio Nacional de Salud, Villa Nueva, Guatemala. [29]National Public Health Reference Laboratory, Georgetown, Guyana. [30]Laboratoire National de Santé Publique, Port-au-Prince, Haiti. [31]Laboratorio Nacional de Vigilancia de la Salud, Tegucigalpa, Honduras. [32]Virology Laboratory, University of the West Indies, Kingston, Jamaica. [33]Instituto de Diagnóstico y Referencia Epidemiologicos, Mexico City, Mexico. [34]Laboratorio Central de Salud Pública, Asuncion, Paraguay. [35]Instituto Nacional de Salud, Chorillos, Peru. [36]Centraal Laboratorium Bureau voor Openbare Gezondheidszorg, Paramaribo, Suriname. [37]The University of the West Indies - St. Augustine, St. Augustine, Trinidad and Tobago. [38]Departamento de Laboratorio de Salud Pública, Montevideo, Uruguay. [39]Instituto Nacional de Higiene Rafael Rangel, Caracas, Venezuela.

