## [Peer Review File · Nature Communications]

Dispersion patterns of SARS-CoV-2 variants Gamma, Lambda and Mu in Latin America and the CaribbeanREVIEWER COMMENTS

Reviewer #1 (Remarks to the Author):

Comments to the authors (manuscript NCOMMS-23-47320)

In this manuscript, the authors describe a study where they investigated the spread of SARS-CoV-2 variants in Latin America and the Caribbean (LAC) and collected several data through COVIGEN network. The manuscript importantly fills a gap in the current literatures of SARS-CoV-2 by analyzing its spread in a region that has been under covered elsewhere, and the manuscript is overall well written. The conclusions are as expected and are very well reflected in the final part of the article.

Minor revisions

- In the present manuscript, there is a strong focus on variants of SARS-CoV-2, however, it is unclear whether the variants in any way impacted each LAC's country transmission dynamics. The connection to shaping the local dynamics should be made more explicit. Also, it is unclear from the manuscript if shifts in country frequencies of variants are due to changes in variant frequencies elsewhere or by chance, or whether these are due to differences in LAC's country specific local transmission dynamics.

- Some sections of the results belong to the method section. I would suggest to take the following sections under the method section. For example :

Section of Strengthening of genomic surveillance capacity through COVID-19 Genomic Surveillance Regional Network (COVIGEN) "As a collaborative networking among PAHO, national authorities and national public health laboratories, COVIGEN implementation was key for strengthening timely SARS-CoV-2 genomic surveillance, facilitating streamlined logistics, procurement and distribution of sequencing reagents, in-country and sub-regional trainings in genomic sequencing and bioinformatics, and for guidance on official notifications through international health regulations. The heterogenic network was based on Nanopore and Illumina technologies and was composed by countries that perform in-house sequencing or external sequencing, held on reference sequencing laboratories

Section of Sampling overview and molecular diversity

To create a comprehensive view of the viral lineage diversity in the continent during the pandemic first two years, we complemented the COVIGEN sampling with all SARS-CoV-2 genomes and metadata from the same period and countries as available in EpiCoV database in GISAID (<https://www.gisaid.org/>).

- Result section should start from description of Figure 1. There should be extensive description of total number of sequenced COVID-19 cases per country vs identified variants as shown in Suppl. Fig. 1.

Reviewer #2 (Remarks to the Author):

Gräf et al. have analysed the transmission patterns of SARS-CoV-2 across countries in Latin America and the Caribbean (LAC). This large scale study would not be possible without the coordinated efforts of international genomic surveillance networks. The work performed here is commendable and as far as I am aware, this is the only paper to trace the migration pattern across South American countries to this scale. Subsampling genomes is always an issue with these kinds of studies, and the authors have implemented a subsampling strategy that has been thought through and explained clearly. The writing is clear and polished. The figures are striking and I appreciate the consistent colour coding used throughout the paper. My feedback is minor.

Main concern: This article provides a large scale overview of the pandemic in LAC. However it should include a more diverse and modern range of references that integrate the results of this paper with the many phylodynamic studies conducted at national levels in LAC, particularly related to the three variants considered here. The authors have covered Brazil quite closely, but other key

countries like Colombia and Peru could do with the same treatment. Are the results from this paper, particularly around migration, congruent with previous studies? Or are there large discrepancies? Some suggested references from 2023:

Justo Arevalo, Santiago, et al. "Phylodynamic of SARS-CoV-2 during the second wave of COVID-19 in Peru." *Nature Communications* 14.1 (2023): 1-13.

Nabaes Jodar, Mercedes Soledad, et al. "The Lambda Variant in Argentina: Analyzing the Evolution and Spread of SARS-CoV-2 Lineage C. 37." *Viruses* 15.6 (2023): 1382.

Sahadeo, Nikita SD, et al. "Correction: Implementation of Genomic Surveillance of SARS-CoV-2 in the Caribbean: Lessons learned for sustainability in resource-limited settings." *PLOS Global Public Health* 3.9 (2023): e0002393.

Jimenez-Silva, Cinthy, et al. "Genomic epidemiology of SARS-CoV-2 variants during the first two years of the pandemic in Colombia." *Communications Medicine* 3.1 (2023): 97.

Orf, Gregory S., et al. "The Principles of SARS-CoV-2 Intervariant Competition Are Exemplified in the Pre-Omicron Era of the Colombian Epidemic." *Microbiology Spectrum* 11.3 (2023): e05346-22.

Minor corrections

Missing reference in introduction: Error! Bookmark not defined.

The authors should make it clear in the introduction what date COVIGEN was founded

Line 87: "important" is a strange word to describe a virus. How about "significant" or "highly transmissible" variants? This will also make the paragraph less repetitive, as the previous sentence also uses the word 'important'

Line 113 and 503: the word continent is used incorrectly. LAC includes countries/territories from two continents: North America and South America, plus there are also some islands (eg. Bermuda) that may or may not have a continent depending on who you ask. I suggest 'continents' or perhaps 'Americas' or 'LAC' or 'continents and surrounding islands'

Fig 1. Caption: What do the circles sizes mean? Number of genomes I assume

Fig 3: I find these figures too hard on the eye, an information overload. It would help if the grid was removed, or if the grid size was larger / lines thinner

Fig 4: No complaints - this is a perfect picture

Line 422: "populations from the South American" -> should this say "South America" or "South American Southern cone" ?

The version of Pango used to make classifications should be specified in methods

Reviewer #3 (Remarks to the Author):

The manuscript by Graf and colleagues explores the genetic diversity of SARS-CoV-2 during the first two years in Latin America and Caribbean regions. Specifically, the authors focus on the reconstruction and the diffusion patterns of three local variants - Gamma, Lambda and Mu. The pivotal role played by COVIGEN in bolstering genomic surveillance capacity in the region should be duly acknowledged. Utilizing cutting-edge phylodynamic methods, the authors illustrate a strong correlation between population-level human mobility data and the inferred migration of the virus.

However, given the widespread nature of the pandemic surely the authors could have expected this. While the manuscript is interesting albeit concise, the completely novelty of the work appears somewhat unclear considering that multiple research groups have demonstrated similar correlation throughout the course of the pandemic.

Others comments:

1. How do the authors deal with the impact of potential sampling bias of their diffusion model?
2. Can the authors measure the relative importance of lineage introduction events and their association with incidence. I am wondering if the authors can dissect the roles of new introductions versus sustained transmissions over the study period?
3. Given the knowledge that these variants are not high in abundance outside South America why did the authors not to choose sequences from Europe and North America as it is very probable that they are obtaining an incomplete picture of the true inferred migration pathways.
4. As illustrated by Tay et al. (MBE, 2022) the VOC are driven by an increase in the substitution rate of around 4-fold the background phylogenetic rate. The authors use a fixed rate of 7.5×10^{-4} substitutions/site/year but I am concerned that the rates for these VOCs is considerably higher. What is the justification for using this fixed rate which may be a bit outdated now compared to the rates estimated by Tay et al.
5. It would be interesting for the authors to elaborate on how the lifting of control measures potentially altered the exchange rate of the virus. Particularly, the lower predictive value of air traffic when modelling the diffusion process of Mu. Do the authors have any data even historical to account for land/water based means of transport to account for the spread of Venezuela and Ecuador. Have the considered that this may also be the result of incomplete sampling?

RESPONSE TO REVIEWERS - Manuscript ID NCOMMS-23-47320

"SARS-CoV-2 genomic surveillance in Latin America and the Caribbean: dispersion patterns of the Lambda, Mu and Gamma variants unveiled"

Reviewer #1

In this manuscript, the authors describe a study where they investigated the spread of SARS-CoV-2 variants in Latin America and the Caribbean (LAC) and collected several data through COVIGEN network. The manuscript importantly fills a gap in the current literatures of SARS-CoV-2 by analyzing its spread in a region that has been under covered elsewhere, and the manuscript is overall well written. The conclusions are as expected and are very well reflected in the final part of the article.

Minor revisions

In the present manuscript, there is a strong focus on variants of SARS-CoV-2, however, it is unclear whether the variants in any way impacted each LAC's country transmission dynamics. The connection to shaping the local dynamics should be made more explicit. Also, it is unclear from the manuscript if shifts in country frequencies of variants are due to changes in variant frequencies elsewhere or by chance, or whether these are due to differences in LAC's country specific local transmission dynamics.

Authors' response:

The connection between the emergence of SARS-CoV-2 variants and the epidemic dynamics was already demonstrated in numerous publications. Focusing on the LAC region, in the introduction section of our work (in the third paragraph), we cited several studies that correlated the emergence of VOCs and VOIs with new COVID-19 epidemic waves. Besides that, our own analyses demonstrated that COVID-19 cases rise soon after the most intense period of variants' inflow (second paragraph of page 5 and Figure 3). This was especially observed for Gamma and Lambda in Chile and Argentina, Gamma in Venezuela, and Mu in Greater Antilles. Moreover, in Figure 2 we can clearly see that new epidemic waves are triggered by new viral variants that emerged locally or were introduced. Still, to stress these conclusions, we modified the discussion section to make more explicit the correlation between variants emergence or introduction and the increasing number of COVID-19 cases.

Some sections of the results belong to the method section. I would suggest to take the following sections under the method section. For example :

Section of Strengthening of genomic surveillance capacity through COVID-19 Genomic Surveillance Regional Network (COVIGEN):

“As a collaborative networking among PAHO, national authorities and national public health laboratories, COVIGEN implementation was key for strengthening timely SARS-CoV-2 genomic surveillance, facilitating streamlined logistics, procurement and distribution of sequencing reagents, in-country and sub-regional trainings in genomic sequencing and bioinformatics, and for guidance on official notifications through international health regulations. The heterogenic network was based on Nanopore and Illumina technologies and was composed by countries that perform in-house sequencing or external sequencing, held on reference sequencing laboratories.”

Section of Sampling overview and molecular diversity:

“To create a comprehensive view of the viral lineage diversity in the continent during the pandemic first two years, we complemented the COVIGEN sampling with all SARS-CoV-2 genomes and metadata from the same period and countries as available in EpiCoV database in GISAID (<https://www.gisaid.org/>).”

Authors' response:

Complying with the reviewer comment, we moved to methods the section describing the COVIGEN implementation. However, since methods are depicted in the end of the study, we believe that a brief explanation of the methods is important to be presented before the results. Then we decided to keep in the results section, the sentence describing how we complemented the COVIGEN data with GISAID data.

Result section should start from description of Figure 1. There should be extensive description of total number of sequenced COVID-19 cases per country vs identified variants as shown in Suppl. Fig. 1.

Authors' response:

We have changed the initial part of the results section and expanded the description of Figure 1.

Reviewer #2

Gräf et al. have analysed the transmission patterns of SARS-CoV-2 across countries in Latin America and the Caribbean (LAC). This large scale study would not be possible without the coordinated efforts of international genomic surveillance networks. The work performed here is commendable and as far as I am aware, this is the only paper to trace the migration pattern across South American countries to this scale. Subsampling genomes is always an issue with these kinds of studies, and the authors have implemented a subsampling strategy that has been thought through and explained clearly. The writing is clear and polished. The figures are striking and I appreciate the consistent colour coding used throughout the paper. My feedback is minor.

Main concern

This article provides a large scale overview of the pandemic in LAC. However it should include a more diverse and modern range of references that integrate the results of this paper with the many phylodynamic studies conducted at national levels in LAC, particularly related to the three variants considered here. The authors have covered Brazil quite closely, but other key countries like Colombia and Peru could do with the same treatment. Are the results from this paper, particularly around migration, congruent with previous studies? Or are there large discrepancies? Some suggested references from 2023:

Justo Arevalo, Santiago, et al. "Phylodynamic of SARS-CoV-2 during the second wave of COVID-19 in Peru." *Nature Communications* 14.1 (2023): 1-13.

Nabaes Jodar, Mercedes Soledad, et al. "The Lambda Variant in Argentina: Analyzing the Evolution and Spread of SARS-CoV-2 Lineage C. 37." *Viruses* 15.6 (2023): 1382.

Sahadeo, Nikita SD, et al. "Correction: Implementation of Genomic Surveillance of SARS-CoV-2 in the Caribbean: Lessons learned for sustainability in resource-limited settings." PLOS Global Public Health 3.9 (2023): e0002393.

Jimenez-Silva, Cinthy, et al. "Genomic epidemiology of SARS-CoV-2 variants during the first two years of the pandemic in Colombia." Communications Medicine 3.1 (2023): 97.

Orf, Gregory S., et al. "The Principles of SARS-CoV-2 Intervariant Competition Are Exemplified in the Pre-Omicron Era of the Colombian Epidemic." Microbiology Spectrum 11.3 (2023): e05346-22.

Authors' response

Previous studies have mainly focused in the within country patterns of dispersion, without a broad international sampling with a rigorous control of sampling bias. Thus, the possibility of comparisons of our results regarding the international variants' spread patterns are limited. Even though, we have expanded our discussion to include the studies suggested by the reviewer.

Minor corrections

Missing reference in introduction: Error! Bookmark not defined.

Authors' response:

We are sorry for this mistake. It was corrected in the revised version.

The authors should make it clear in the introduction what date COVIGEN was founded

Authors' response:

Done

Line 87: "important" is a strange word to describe a virus. How about "significant" or "highly transmissible" variants? This will also make the paragraph less repetitive, as the previous sentence also uses the word 'important'

Authors' response:

Done

Line 113 and 503: the word continent is used incorrectly. LAC includes countries/territories from two continents: North America and South America, plus there are also some islands (eg. Bermuda) that may or may not have a continent depending on who you ask. I suggest 'continents' or perhaps 'Americas' or 'LAC' or 'continents and surrounding islands'

Authors' response:

Done

Fig 1. Caption: What do the circles sizes mean? Number of genomes I assume

Authors' response:

The information was added to the figure legend.

Fig 3: I find these figures too hard on the eye, an information overload. It would help if the grid was removed, or if the grid size was larger / lines thinner

Authors' response:

We redo the figure with larger grid size.

Fig 4: No complaints - this is a perfect picture

Authors' response:

Thank you!

Line 422: "populations from the South American" -> should this say "South America" or "South American Southern cone" ?

Authors' response:

Done

The version of Pango used to make classifications should be specified in methods

Authors' response:

The SARS-CoV-2 lineage classification used in this study was the one present in the genome metadata file supplied by GISAID when the data was downloaded. Unfortunately, the Pango version used to classify the genomes is not informed in the GISAID metadata file.

Reviewer #3

The manuscript by Graf and colleagues explores the genetic diversity of SARS-CoV-2 during the first two years in Latin America and Caribbean regions. Specifically, the authors focus on the reconstruction and the diffusion patterns of three local variants - Gamma, Lambda and Mu. The pivotal role played by COVIGEN in bolstering genomic surveillance capacity in the region should be duly acknowledged. Utilizing cutting-edge phylodynamic methods, the authors illustrate a strong correlation between population-level human mobility data and the inferred migration of the virus. However, given the widespread nature of the pandemic surely the authors could have expected this. While the manuscript is interesting albeit concise, the completely novelty of the work appears somewhat unclear considering that multiple research groups have demonstrated similar correlation throughout the course of the pandemic.

Other comments:

1. How do the authors deal with the impact of potential sampling bias of their diffusion model?

Authors' response:

To reduce sampling disparities among locations, we subsampled each variant dataset proportionally to the cumulative number of COVID-19 cases attributable to Gamma, Lambda and Mu in each location. An arbitrary number of 1 sequence / 10,000 Gamma attributable COVID-19 cases and 1 sequence / 1,000 Lambda or Mu attributable COVID-19 cases were selected, with a minimum number of 100 sequences (or the maximum available when less than that) per location. To maximize the temporal coverage, sequences were grouped by epidemiological week and sampled as evenly as possible in each location. More details about our approach to reduce sampling bias is provided in Methods section (second paragraph of page 13) and in the Discussion section (second paragraph of page 11), we added a sentence highlighting this.

2. Can the authors measure the relative importance of lineage introduction events and their association with incidence. I am wondering if the authors can dissect the roles of new introductions versus sustained transmissions over the study period?

Authors' response:

In Figure 3 we co-plotted the number of cases attributable by each variant and the ratio of variant specific viral outflow over total viral flow. Although this analysis doesn't present a formal statistical association test, we can visually observe that COVID-19 cases rise soon after the most intense period of variants' inflow (second paragraph of page 5). This was especially

observed for Gamma and Lambda in Chile and Argentina, Gamma in Venezuela, and Mu in Greater Antilles. The outcome of a formal association test between viral introductions and incidence of cases would be challenging to interpret because incidence is the product of introductions and local transmission, and their dynamics can fluctuate over time. Moreover, as observed in Figure 3, the temporal gap between a variant introduction and an increase in incidence can vary greatly between countries, depending on local social distance measures that can deeply impact the success of variant spread. As an attempt to measure the success of each variant introduction, we analyzed the proportion of descendants originated from the introductions. Thus, the relative importance each introduction was measured, to a large extent, by the number of transmission chains (a percentage of all transmissions in the phylogeny) that were originated from each location. These results are presented in the last paragraph of page 7 and are further discussed in the second paragraph of page 9.

3. Given the knowledge that these variants are not high in abundance outside South America why did the authors not to choose sequences from Europe and North America as it is very probable that they are obtaining an incomplete picture of the true inferred migration pathways.

Authors' response:

Gamma, Lambda and Mu represented a very small fraction (less than 1%) of the variants circulating in other regions than LAC in the period of time analyzed in this study. Adding countries outside LAC in our analyses, even if they contributed with few genomes, would drastically increase the number of locations in our diffusion model, which would incur in MCMC convergence issues. Moreover, including countries outside LAC would not add interesting information for our main porpoise in this study which was to reconstruct the SARS-CoV-2 variants migration patterns within the region. We would very likely see several exportations of VOC Gamma and VOIs Lambda and Mu to overseas, but we would unlikely see a movement in the reverse direction. The only exception could be some exchange of Gamma lineages between Mexico and USA, but we have already acknowledged this caveat in the Discussion section (second paragraph of page 11).

4. As illustrated by Tay et al. (MBE, 2022) the VOC are driven by an increase in the substitution rate of around 4-fold the background phylogenetic rate. The authors use a fixed rate of 7.5×10^{-4} substitutions/site/year but I am concerned that the rates for these VOCs is considerably higher. What is the justification for using this fixed rate which may be a bit outdated now compared to the rates estimated by Tay et al.

Authors' response:

Tay et al. (<https://doi.org/10.1093/molbev/msac013>) have demonstrated an episodic increase in the substitution rate during the process of VOC emergence. This short acceleration in the SARS-CoV-2 evolution can explain how dozens of mutations are quickly fixed in the viral population in a short time, likely due to strong selective pressures. However, this accelerated substitution rate lasts only during the process of variant emergence, in the stem branch that leads to the variant clade. When all important mutations that define the variant are fixed, the rate slowdown to values close to what we used in our analyses, as said in Tay et al.: "The mean substitution rate of branches other than the VOC stems was 0.65×10^{-3} subs/site/year (95% CI: $0.58-0.77 \times 10^{-3}$) in the UCLN and 0.69×10^{-3} subs/site/year (95% CI: $0.60-0.80 \times 10^{-3}$) for the UCG". This behavior of SARS-CoV-2 evolution was also demonstrated in our previous work about Gamma origins in Brazil (Gräf et al., 2021 - <https://doi.org/10.1093/ve/veab091>). In the current work, we included in the analyses only Gamma, Lambda and Mu genomes with all defining mutations, thus skipping the period of increased substitution rate. A fixed substitution rate was applied to relieve computational burden and accelerate MCMC convergence, because we were analyzing datasets of thousands of genomes. This strategy was

proposed by Dellicour et al. 2020 (<https://doi.org/10.1093/molbev/msaa284>) as a workflow to analyze big datasets.

5. It would be interesting for the authors to elaborate on how the lifting of control measures potentially altered the exchange rate of the virus. Particularly, the lower predictive value of air traffic when modelling the diffusion process of Mu. Do the authors have any data even historical to account for land/water based means of transport to account for the spread of Venezuela and Ecuador. Have the considered that this may also be the result of incomplete sampling?

Authors' response:

A direct assessment of alternative (besides airplanes) modes of transportation is challenging due to the lack of detailed data on its routes, number of passengers and because the data is not centralized and curated. Thus, we did not include such data in our study. We found that Mu mainly spread from Colombia to Central America and the Caribbean, in a process poorly explained by air traffic fluxes. Another study investigating the initial chikungunya virus spread in the Caribbean islands found that spatial transmission was better described by geographic proximity than by air transportation fluxes (10.2807/1560-7917.ES2014.19.28.20854), which is very likely explained by the movement of individuals by boat. This adds context to our findings and highlights the importance of maritime transport in the Caribbean. We have added this in the discussion section to support our findings.

REVIEWERS' COMMENTS

Reviewer #3 (Remarks to the Author):

The authors have addressed all my prior concerns.